# Mechanisms of Immune-Related Long Non-Coding RNAs in Spleens of Mice Vaccinated with 23-Valent Pneumococcal Polysaccharide Vaccine (PPV23)

**DOI:** 10.3390/vaccines11030529

**Published:** 2023-02-23

**Authors:** Nan Zhu, Fan Zhang, Huan Zhou, Wei Ma, Haiguang Mao, Mengting Wang, Zhijian Ke, Jinbo Wang, Lili Qi

**Affiliations:** 1School of Biological and Chemical Engineering, NingboTech University, Qianhunan Road 1, Ningbo 315100, China; 2Aimei Vacin BioPharm (Zhejiang) Co., Ltd., Ningbo 315000, China

**Keywords:** PPV23, lncRNA, mRNA, spleen, immune

## Abstract

The 23-valent pneumococcal vaccine (PPV23) is a classical common vaccine used to prevent pneumococcal disease. In past decades, it was thought that vaccination with this vaccine induces humoral immunity, thereby reducing the disease associated with infection with 23 common serotypes of Streptococcus pneumoniae (Sp). However, for this polysaccharide vaccine, the mechanism of immune response at the transcriptional level has not been fully studied. To identify the lncRNAs (long noncoding RNAs) and mRNAs in spleens related to immunity after PPV23 vaccination in mice, high-throughput RNA sequencing of spleens between a PPV23 treatment group and a control group were performed and evaluated in this study. The RNA-seq results identified a total of 41,321 mRNAs and 34,375 lncRNAs, including 55 significantly differentially expressed (DE) mRNAs and 389 DE lncRNAs (*p* < 0.05) between the two groups. GO and KEGG annotation analysis indicated that the target genes of DE lncRNAs and DE mRNAs were related to T-cell costimulation, positive regulation of alpha–beta T-cell differentiation, the CD86 biosynthetic process, and the PI3K-Akt signaling pathway, indicating that the polysaccharide component antigens of PPV23 might activate a cellular immune response during the PPV23 immunization process. Moreover, we found that Trim35 (tripartite motif containing 35), a target gene of lncRNA MSTRG.9127, was involved in regulating immunity. Our study provides a catalog of lncRNAs and mRNAs associated with immune cells’ proliferation and differentiation, and they deserve further study to deepen the understanding of the biological processes in the regulation of PPV23 during humoral immunity and cellular immunity.

## 1. Introduction

The 23-valent pneumococcal vaccine (PPV23) is a classical common vaccine used to prevent pneumococcal disease (PD) [1]. Streptococcus pneumoniae (Sp), as one of the common pathogens, can cause pneumonia, meningitis, bronchitis, and other diseases through droplets [2]. The bacterium is Gram-positive and was first isolated from the sputum of a patient in 1881, and PD caused by Sp is one of the major causes of death among young children and the elderly [3]. In recent years, there have been increasing reports of Sp resistance, which brings difficulties to the treatment of pneumonia [4,5]. Therefore, vaccination is the most cost-effective option.

PPV23 is a clear, sterile liquid made by mixing the purified capsules of various serotypes of Sp with sodium chloride and other excipients [6]. Most previous studies reported that because capsular polysaccharide (CPS) is a non-T-cell-dependent antigen, it could not produce immune memory and produced IgM antibodies only, which had a short duration of immunity [7]. Many previous studies have shown that the pneumococcal polysaccharide vaccine can induce non-T-cell-dependent humoral immune response, so it is more suitable for the susceptible population over 2 years old or over 50 years old [8]. It was reported that children younger than 2 years of age had a much higher incidence than other age groups, with 75% of cases occurring in this group, followed by the elderly with chronic obstructive pulmonary disease (COPD) and asthma [3,9]. The capsule is a gelatinous substance located on the outside of the cell wall of Sp, and its main component is polysaccharide [10]. Therefore, the capsule is an important structural substance for bacterial growth and reproduction [11]. CPS is an important virulence factor of Sp. It has strong immunogenicity and is often used as one of the target antigens for vaccine preparation because it has fewer structural changes [12]. As the main virulence factor of Sp, CPS is closely related to its serotype; specifically, different serotypes produce different virulence factors [6].

Long noncoding RNAs (lncRNAs) are from the regions of the transcriptome with lengths longer than 200 nucleotides and without an evident protein-coding capacity [13]. A number of previous studies proved that the lncRNAs play key roles in the regulation of mRNA expression by directly affecting the transcription process or by recruiting epigenetic complexes [14,15]. Due to the molecular mechanisms of lncRNAs, they can also act as sponges, guides, decoys, signals, and scaffolds during the biological process [16]. LncRNAs can specifically combine with certain mRNAs to interfere with their splicing activity and change the structure of proteins by directly binding to enzymes and changing the catalytic activities through an allosteric effect [17]. In some cases, transcription and splicing of lncRNAs are necessary to induce the expression of nearby protein-coding genes, and the lncRNA locus might contain functional elements such as enhancers embedded within it that might be key to the expression of the nearby protein-coding gene [18]. Nuclear lncRNAs are typically associated with chromatin modification, RNA processing, and transcriptional regulation; cytoplasmic lncRNAs are usually associated with mRNA stability or translation, and act as direct agonists or antagonists of protein expression [19]. As a consequence, the genetic molecular mechanisms of cell proliferation and differentiation, regulation of the cell cycle, dosage compensation, and epigenetics all participate in the protein inhibition by miRNA titration or by binding of lncRNAs to proteins or to miRNAs [20]. The molecular mechanisms of lncRNAs allow them to act as guides, decoys, signals, and scaffolds during the biological process [16]. It was reported that lncRNAs are important regulators in innate immunity against infection. They are involved in numerous aspects of innate immune response, including the macrophages and dendritic cells, the differentiation and apoptosis of myeloid cells and the activation of monocytes, and the maintenance of hematopoietic stem cells [21,22]. LncRNAs can interact with the core site of MLL H3K4 methyltransferase to promote histone methylation at the Ifng site of CD8+/Th1 cells [23]. Compared with T cells, little is known about the function of lncRNAs in B cells; so far Bolland et al. [24] explained the role of lncRNAs in chromatin remodeling, which was associated with variable, diverse, and linked (V/D/J) gene recombination required to produce antigen receptors (TCR or Ig). Although lncRNAs have been identified as key regulatory factors in a number of biological processes including immunity, the regulatory mechanisms of most lncRNAs in immune processes still remain largely unknown [25]. Despite the rapid increase in data related to lncRNAs, little is known about their exact function, their mechanism of action, or even how many different types of lncRNAs exist. Although lncRNAs generally exhibit poor evolutionary conservation [26], it is clear that they play an important role in immunity through various biological pathways, including the regulation of immune responses after vaccination. However, until now, whether lncRNAs play a role in PPV23 immunization, which lncRNA plays a key role, and the exact immune mechanism, remained unclear. Therefore, in this study, high-throughput transcriptome sequencing technology was used to preliminarily explore the molecular mechanism of lncRNA in the immune process of PPV23, so as to lay a certain research foundation for increasing the understanding of the mechanism of action of PPV23.

In the present study, in order to identify the lncRNAs and mRNAs in spleens related to immunity in mice after PPV23 vaccination, high-throughput RNA sequencing of spleens between the PPV23 treatment group and the control group were performed. The expression of lncRNA transcriptome in the spleens of mice was analyzed to preliminarily explore the effect and mechanism of these lncRNAs on the immune response. It provides novel ideas and references for simulating the new immune antigen scheme for disease treatment in the future.

## 2. Materials and Methods

### 2.1. Animal Treatment and Sample Collection

A total of 24 four-week-old male ICR mice with the same genetic background were randomly divided into 2 groups (PPV23 group and control group) with 12 mice in each group. Each cage housed 3 mice, and all experiment animals were given free access to food and water. All the experiment mice were commissioned to the Laboratory Animal Center of Zhejiang University, with the temperature of the mouse rearing house 23 ± 1 ℃ and the humidity 50–60%. In addition, all animal procedures were approved by the Animal Welfare Committee of Zhejiang University.

The PPV23 vaccination procedure is as follows: The mice were kept quiet for three days to acclimate to the new environment, and then vaccinated on the third day of the experiment. The amount of vaccine given was based on the weight of the mice. The PPV23 was applied by *Aimei Vacin BioPharm (Zhejiang) Co., Ltd.*, and the vaccine was injected intramuscularly into the lateral thigh muscle of the mice. Serum and spleen tissues were collected two weeks after injection. The spleen tissues were collected and immediately frozen in liquid nitrogen to isolate RNA. The serum was isolated and tested for antibody concentration (IgG, IgM, IgA) immediately with ELISA.

### 2.2. RNA Isolation, Library Preparation, and Sequencing

The total RNA of the mice spleens was isolated and purified with TRIzol reagent (Invitrogen, Carlsbad, CA, USA) according to the kit instructions. The RNA amount and purity of the spleen samples were measured with the NanoDrop ND-2000C (Thermo, Wilmington, DE, USA). The integrity of the extracted RNA was measured using Agilent 2100 with RIN number > 7.0. About 5 μg of extracted total RNA were taken for the experiment to remove rRNA using the procedure of the Ribo-Zero™ rRNA Removal Kit (Illumina, San Diego, CA, USA). Then the rest of the RNAs were fragmented into many small pieces with divalent cations at high temperature after depleting rRNAs. The cleaved RNA fragments were subsequently reversed into cDNA; then they were used to synthesize the U-labeled second-stranded DNAs with dUTP, E. coli DNA polymerase I, and RNase H. The average insert size of the final cDNA library was approximately 300 bp. In the end, Illumina Hiseq 4000 (LC Bio, Hangzhou, China) was applied for the paired-end sequencing following the recommended instruction of the apparatus.

### 2.3. Quality Control and Mapping

The low-quality reads of the sequencing data, including low-quality bases, undetermined bases, and adaptor contamination, were removed with Cutadapt. FastQC was applied to verify the sequence quality. Hisat2 [27] and Bowtie2 [28] were then used to map the obtained reads to the genome of the mouse in NCBI (*Mus musculus*, assembly GRCm39). StringTie [29] was applied to assemble the mapped reads. All the transcripts from the mouse spleens were combined to reconstruct a comprehensive transcriptome using a Perl script. Ballgown [30] and StringTie [29] were applied to estimate the expression levels of all the transcripts after the final transcriptomes were generated.

### 2.4. Identification of lncRNAs

At first, transcripts overlapped with known mRNAs or shorter than 200 bp were deleted. Transcripts with coding potential were predicted with CNCI [31] and CPC [32]. The transcripts with CNCI scores < 0 and CPC score < −1 were discarded. The remaining transcripts were considered to be lncRNAs.

### 2.5. Analysis of DE mRNAs and lncRNAs

StringTie was applied to calculate the mRNA and lncRNA expression levels with FPKM. R package-Ballgown was applied to select the DE mRNAs and DE lncRNAs (log2 (fold change) <−1 or log2 (fold change) >1 and with *p* value < 0.05). The significance threshold was from FDR-based adjusted *p*-values.

### 2.6. Target Gene Prediction and Functional Analysis of lncRNAs

We predicted the cis-target genes of lncRNAs to explore the function of the identified lncRNAs, which were able to play a cis role on the neighboring target genes in the mice spleens. In the current study, the coding genes in 100 kb downstream and upstream were selected using Python script. At last, the BLAST2GO was applied to perform the functional analysis of the target genes [33]. The Pearson Correlation was then calculated by the normalized expression values (FPKM), and the correlation threshold *p* value < 0.05 was used to select lncRNA-mRNA co-expressed pairs.

### 2.7. GO and KEGG Enrichment Analysis

In order to better understand the biological functions of DE mRNA and DE lncRNAs in mice spleens treated with PPV23, Gene Ontology (GO) terms and Kyoto Encyclopedia of Genes and Genomes (KEGG) pathway enrichment analysis were carried out to further explore the biological processes. GO terms were performed with BLAST2GO with the significance *p* value < 0.05.

### 2.8. RNA-Seq Result Validation by qRT-PCR

Six lncRNAs (MSTRG.26454.3, MSTRG.27119.1, MSTRG.25551.7, MSTRG.12225.1, MSTRG.12876.1, and MSTRG.21874.1) and six mRNAs (H2ac12, Apold1, Igf2, Myh1, Spp1, and Actn3), which represented the differential expression levels of RNA-seq from the six sequenced mice spleens were selected to run qRT-PCR. The primers used in the current study are shown in Appendix A. ABI Step One Plus System (Applied BiosystemCarlsbad, Foster City, CA, USA) was used to perform qRT-PCR by SYBR Premix Ex Taq Kit (TaKaRa, Dalian, China). Relative gene expression levels were quantified and normalized by mouse *GAPDH* gene using 2^−ΔΔCt^ method. Three independent biological replicates were applied. All of the measurements were repeated in triplicate.

## 3. Results

### 3.1. Phenotypic Data Analysis

Four phenotypic traits were collected, including live weight of the mice and the IgG, IgM, and IgA concentrations in serum (*n* = 12). As shown in Figure 1A, no significant difference (*p* > 0.05) was shown between the body weight of control group and the PPV23 group. Equally unsurprisingly, the PPV23 treatment group showed significantly higher (*p* < 0.01) IgG, IgM, and IgA concentrations in the serum (Figure 1B–D).

### 3.2. Sequencing Data Summary

A total of 83.22 Gb raw data was obtained from 6 libraries. In detail, 86669110, 87116008, and 88966556 raw reads were generated from the control group (Control_1, Control_2, Control_3); 89358780, 85088936, and 82610828 raw reads were generated from the treatment group (PPV_1, PPV_2, PPV_3). All the raw reads were filtered to obtain the clean reads, which were mapped to the *Mus musculus* (assembly GRCm39) version of the mouse genome sequence, with the mapping ratio ranging from 91.82% to 94.38%. The detailed data are show in Appendix A.

### 3.3. Identification of lncRNAs and mRNAs in Mouse Spleens

As shown in Appendix A, a total of 34,375 putative lncRNAs were identified from the six libraries, including 28,108 novel lncRNAs and 6267 known lncRNAs. Regarding the genomic locations of the novel lncRNAs, 3049 were intronic (48.65%), 942 were bidirectional (15.03%), 574 were sense (9.16%), 1273 were intergenic (20.31%), and 429 were antisense lncRNAs (6.85%).

In this study, the average length of the identified lncRNA transcripts is 2656 bp, which shows shorter than 4591 bp length of the mRNA transcripts (Figure 2A). In addition, the number of exons of lncRNAs is 3.60 on average, which is less than that of mRNAs (9.78 on average).

As shown in Figure 2B, 86.02% of lncRNAs contain five or fewer exons, while 69.34% of mRNAs contain four or more exons. In addition, lncRNAs obtained in the current study had shorter open reading frames (ORFs) than mRNAs of spleen tissues in mice (Figure 2C,D).

### 3.4. Identification of DE mRNAs and DE lncRNAs

In order to identify the mRNAs and DE lncRNAs between the control group and PPV23 group, we calculated the DE mRNAs and DE lncRNAs with FPKM levels in the mouse spleens. Figure 3A shows that the lncRNA expression levels were higher than the mRNA expression levels in this study, while Figure 3B shows that the number of lncRNAs was less than that of mRNAs.

A total of 55 DE mRNAs (Appendix A) and 389 DE lncRNAs (Appendix A) were identified between the control group and the PPV23 group. Compared with the control group, 12 mRNAs and 193 lncRNAs were significantly upregulated, while 43 mRNAs and 196 lncRNAs were downregulated. The volcano plots of the DE mRNAs and DE lncRNAs are shown in Figure 4A,B.

### 3.5. Functional Enrichment of DE mRNAs

Gene Ontology (GO) was performed to analyze the main functions of the obtained DE mRNAs. A total of 626 GO terms with functional annotation information were enriched for 55 DE mRNAs. As shown in Appendix A, 307 GO terms were significantly (*p* < 0.05) enriched in the GO analysis results of DE mRNAs. As shown in Figure 5A,B, the significantly enriched GO terms of DE mRNAs were involved in ossification, extracellular region, extracellular space, biomineral tissue development, and osteoblast differentiation. KEGG pathway analysis (Figure 5C) showed nine significantly (*p* < 0.05) enriched pathways, such as ECM-receptor interaction, focal adhesion, PI3K-Akt signaling pathway, human papillomavirus infection, and protein digestion and absorption. The detailed information is shown in Appendix A.

### 3.6. Cis-Regulatory Roles of DE lncRNAs in Spleen Tissues of Mice

To further investigate the regulatory functions of the lncRNAs in the spleen tissues of mice, we forecast the cis-regulated target genes of the differently expressed lncRNAs between the control group and PPV23 group. In the current study, 55 potential lncRNA target genes were found with 100 kb as the cutoff (Appendix A). As shown in Appendix A, GO analysis revealed 99 significant (*p* < 0.05) GO terms based on the cis-regulated target genes. The differentially expressed lncRNA target genes were found to be related with biological processes, including negative regulation of Rac protein signal transduction, paraxial mesoderm formation, negative regulation of fibroblast growth factor production, and lung-associated mesenchyme development. The main molecular function and cellular component categories were related to cyclin-dependent protein kinase activity, lytic vacuole, carbonyl reductase (NADPH) activity, and nodal binding (Figure 6A,B). The KEGG analysis of DE lncRNAs revealed that the target genes of those lncRNAs were mainly enriched in tuberculosis, the C-type lectin receptor signaling pathway, and asthma (Figure 6C, Appendix A). Based on the prediction of DE lncRNA-gene pairs in cis-regulation, the first four and the last four lncRNA-gene pairs were listed in Table 1 by the Pearson correlation coefficient, and the regulation directions of the first four lncRNA-gene pairs showed the same, while the last four pairs were the opposite.

### 3.7. Co-Enriched GO Terms of DE lncRNAs and mRNAs

In order to investigate the crucial pathways of PPV23 to the mouse spleens, a total of five significantly enriched GO terms were identified in both DE lncRNA target gene enrichment and DE mRNA enrichment (Table 2). The significantly co-enriched GO terms were involved in CD86 biosynthetic process, CCR chemokine receptor binding, positive regulation of type IIa hypersensitivity, mRNA cleavage factor complex, and natural killer-cell-mediated cytotoxicity biological, of which three pathways were involved in the cellular component, one involved in the molecular function, and one involved in the biological process, respectively.

### 3.8. DE lncRNAs and DE mRNAs Validation by qRT-PCR

Six DE lncRNAs and six DE mRNAs were selected at random to validate the RNA-seq result by qRT-PCR. The relative fold changes of expression levels performed with qRT-PCR were consistent with the results of RNA-seq data (Figure 7), suggesting that the identification of transcripts and estimation of abundance were highly credible in this study. In addition, the mRNA expression levels of Trim35 gene in PPV23 group showed significantly higher (*p* < 0.01) than those in control group (Figure 8).

## 4. Discussion

PPV23 is one of the most important means to prevent Sp infection [1]. Antibiotics are the treatment of choice for human diseases caused by Sp, but to avoid the overuse of antibiotics, the most aggressive response to such diseases is vaccination. Vaccination can not only improve the utilization efficiency of medical resources under limited conditions, but also greatly reduce the damage caused by unnecessary diseases [4,5]. Currently, Sp vaccines are mainly polysaccharide vaccines, based on the CPS of Sp, and polysaccharide-protein conjugate vaccines. PPV23 is a polysaccharide vaccine, which is a clear sterile liquid made by mixing the purified capsules of various serotypes of Sp [6]. Most previous studies reported that because CPS is a non-T-cell-dependent antigen, it cannot produce immune memory and produces IgM antibodies only, which have a short duration of immunity [7]. Although the PPV23 has been in use for several decades, the specific immune response mechanism is still not fully understood, especially at the transcription levels.

The spleen is an important peripheral immune organ of the animal body, as well as the center of cellular and humoral immunity. When blood flows through the spleen, the spleen recognizes antigens and pathogens, and stimulating molecules in the blood stimulate various receptors in the cells of the spleen, thereby activating the innate immune response [34]. The pattern recognition receptors (PRRs) on various receptor cells are correlated with the immune response pattern of the spleen [35]. Therefore, in this study, the spleen, the largest immune organ, was selected as the research object, and high-throughput RNA sequencing technology was used to analyze the transcription levels of spleens in PPV23-vaccinated mice. We identified a total of 55 DE mRNAs and 389 DE lncRNAs between the PPV23 group and the control group.

In the last decade, a number of researchers had reported that lncRNAs played important roles in various biological activities, including the immunologic process [25]. The current study is the first to report the transcriptome profiling of lncRNAs and mRNA in mouse spleens vaccinated with PPV23. The sequencing data analysis revealed that the lncRNAs identified in this study showed shorter transcript lengths and fewer exons than mRNA, which is consistent with other tissues among different species reported by previous studies [15], indicating that the lncRNA sequencing result identified in this study is reliable. The RNA sequencing result showed that 41.75% of identified lncRNAs in this study were shorter than 1000 bp, while only 21.83% mRNAs were shorter than 1000 bp. Additionally, the average expression levels of identified lncRNAs in this study were much higher (*p* < 0.05) than those of mRNAs in the mouse spleens, implying that the lncRNAs in mouse spleens might play important roles in immunity.

Most previous studies reported that the expression of lncRNAs could regulate the expression levels of the neighboring mRNAs through coactivation or repression and was high correlated with it [18,36]. Consequently, we inferred that there is a molecular mechanism through which lncRNAs could obviously influence the immune process by mediating the putative regulation of the corresponding target mRNAs in mouse spleens. In our study, the DE cis-target genes, which are located within 100 kb upstream and downstream of the identified 389 DE lncRNAs, were obtained to predict the potential biological functions in the assumptive regulation of immunity in mouse spleens. The analysis result suggests that the DE coding gene, named Trim35, might be regulated by the DE lncRNA MSTRG.9127, and Trim35 was significantly up-regulated in the PPV23 treatment group.

Tripartite motif 35 (Trim35), a ubiquitin E3 ligase, is one member of the tripartite motif protein family and has been recognized to play an important role in the immune process [37]. It mediates interferon-beta production by regulating ubiquitination of various adaptor proteins in innate immune signaling pathways [38]. Inhibition of or deficiency in Trim35 suppressed the production of type I interferon (IFN-I) in immune response, and Trim35 gene deficient mice were more susceptible to influenza virus infection than wild-type mice [39]. Meanwhile, it was reported that Trim35 was a negative feedback regulator of TLR7/9-mediated IFN-I production on account of its ability to suppress the stability of IRF7 (the master regulator of type I IFN response) [40]. Our result showed that the mRNA expression levels of the Trim35 gene in the PPV23 group were significantly higher (*p* < 0.01) than those in the control group; thus, we speculated that Trim35 might influence the immune process in mouse spleens by regulating the transcriptional activity of IFN.

Many previous studies have confirmed that multifarious signaling pathways and regulatory mechanisms are involved in the regulation of immunity [25]. In the present study, GO terms and KEGG pathways analyses were performed to further explore the biological functions of the target genes of identified immune-related DE mRNAs and DE lncRNAs in mouse spleens. The result revealed that ECM-receptor interaction and focal adhesion pathways were significantly (*p* < 0.05) enriched. The extracellular matrix (ECM) can interact with cells to provide important signals to control cell function and to maintain homeostasis in vivo, and ECM proteins involved in the matrix organization might be involved in the pathogenesis of pulmonary diseases [41]. Moreover, a number of studies have shown that the focal adhesion pathway participates in airway smooth muscle cell proliferation and repairing bronchial epithelium integrity in pulmonary diseases such as asthma and COPD [42,43]. This evidence indicates that the DE mRNA and DE lncRNA found in this study might relate to the immune response in pulmonary diseases. Interestingly, the analysis results also demonstrated that the DE mRNAs and DE lncRNAs identified in this study participate in the negative regulation of CD4-positive, T-cell costimulation, positive regulation of alpha–beta T-cell differentiation, and so on.

Over the past few decades, most capsular polysaccharides of bacteria were considered to be T-cell-independent antigens that cannot induce helper T-cell activation or stimulate Ig class conversion or immune memory in B cells and are unable to be processed and presented by major histocompatibility complex (MHC) molecules [7,12]. As a result, it was thought that polysaccharide vaccines such as PPV23 could activate humoral immunity only, and the cells responsible for humoral immunity are B cells [7]. To our surprise, the GO terms and KEGG pathways analysis results in this study found several items related to T cells, and this result was similar to several studies reported in recent years [44,45]. With the study of the biological function and immunomodulatory effect of bacterial polysaccharide, more and more of the literature has reported that CPS can activate the T cell population; promote the development of T cells, DCs, or other immune cells; and enhance the functions of immune cells and promote their maturation [7,12]. *B. fragilis* can express eight kinds of capsular polysaccharides (A–H), among which capsular polysaccharide A (PSA) has the strongest immunoregulatory ability [46]. The tetrasaccharide repeating unit of PSA is the zwitterionic polysaccharide, containing a negatively and a positively charged motif in each repeating unit. This special capsular polysaccharide structure enables PSA to directly act as a T-cell antigen to regulate the action of T cells [45]. PSA-induced T-cell activation must be processed by antigen-presenting cells (APCs), which present a polysaccharide (similar to an antigen-presenting peptide) to a major histocompatibility complex Class Ⅱ (MHC-Ⅱ) molecule. The polysaccharide interacts with an α-β T-cell receptor (TCR) on T cells to stimulate the production of IL-10 by CD4+T cells [47]. In addition, PSA could bind to the Toll-like receptor 2 (TLR2) on T cells and directly act on FoxP3+ regulatory T cells (Tregs) [46]. Studies of helper T cell 2 (Th2) in germ-free mice have shown that PSA acts on helper T cell 1(Th1), inducing the host to establish an appropriate Th1/Th2 balance to promote intestinal or splenic homeostasis and immune system development [48,49]. Moreover, PSA has now been shown to activate T-cell populations and prevent various inflammatory diseases by inhibiting pathogenic inflammatory cells [50]. PSA reduces brain stem inflammation and prevents fatal herpes simplex encephalitis (HSE) by binding to enteric-resident plasma blasts and inducing secretion of IL-10 and gamma-interferon (IFNγ) by CD4+T cells and CD8+T cells [51,52]. It was reported that oral PSA or PSA transfer immunized mouse spleen T cells could prevent leukocyte infiltration and lung lesions before inducing trachea inflammation, thus preventing asthma [8]. Furthermore, PSA induces the production of IL-10 by FoxP3+ regulatory T cells [53]. The previous studies reported that all polysaccharides belong to T-cell-independent type 2 (TI-2) antigens. After entering the body, they bind to antigen recognition receptors on the surface of B cells, namely, surface immunoglobulin molecules, crosslinked and activated B cells, causing a series of downstream changes, so that B cells differentiate into plasma cells, which can secrete antibodies [7]. However, with the above evidence combined with the results of this study, we speculate that T cells might be involved in the immunization process of PPV23. The specific molecular mechanism of how T cells are involved in the immune process caused by CPS of PPV23 still needs further identification and intensive study.

## 5. Conclusions

Above all, this study is the first comprehensive description of mRNA and lncRNA profiles of mice spleens treated with PPV23, and several DE mRNAs and DE lncRNAs were predicted to be associated with immunity in mice spleens after PPV23 vaccination. Additionally, the DE mRNAs and DE lncRNAs identified in the current study could provide new insights for further understanding the mechanism of immunity by PPV23. The lncRNA MSTRG.9127 might play an important regulatory role in the immunologic process treated with PPV23 by affecting its potential target gene, Trim35. The CPS antigen of polysaccharide vaccines participates in the body’s immunity through lncRNAs, and its immune mechanism is likely to be related to T cells. Therefore, our study lays a foundation for understanding the immune mechanism of polysaccharide vaccines and the development of future vaccines. More molecular and cellular experiments should be carried out to verify the sequencing data.

## Figures and Tables

**Figure 1 vaccines-11-00529-f001:**
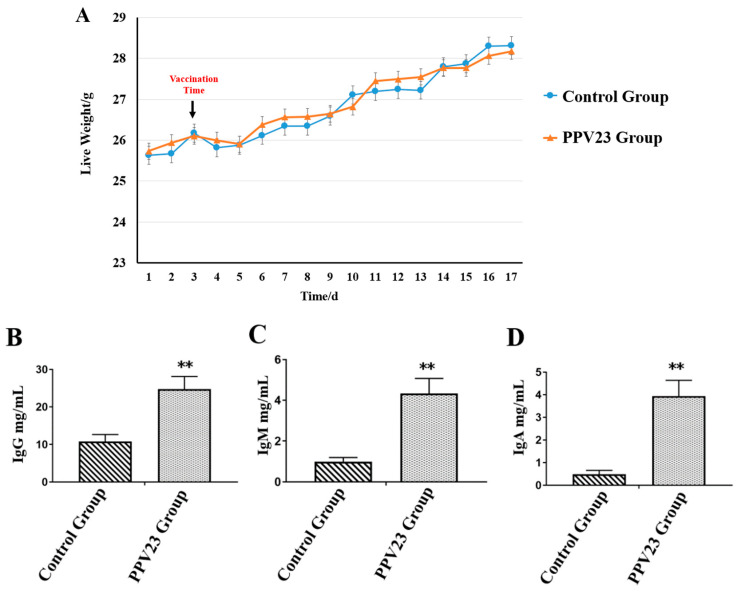
Phenotypic data analysis of mice between control group and PPV23 treatment group (n = 12). (**A**) Analysis of mice live weight during the experiment periods. (**B**) Serum immune factor concentration of IgG. (**C**) Serum immune factor concentration of IgM. (**D**) Serum immune factor concentration of IgA. ** indicates significant differences (*p* < 0.01).

**Figure 2 vaccines-11-00529-f002:**
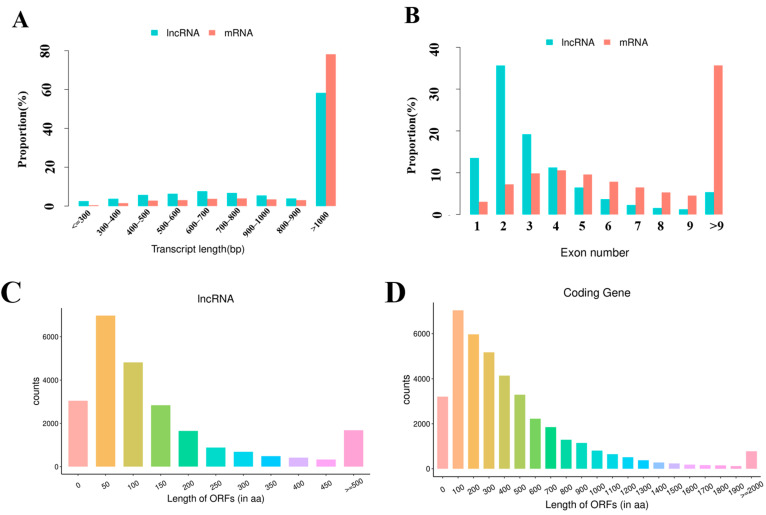
Genomic features of lncRNAs in mouse spleens. (**A**) The transcript length distribution of lncRNAs and mRNAs. (**B**) The exon number distribution of lncRNAs and mRNAs. (**C**) The ORFs length distribution of lncRNAs. (**D**) The ORFs length distribution of mRNAs.

**Figure 3 vaccines-11-00529-f003:**
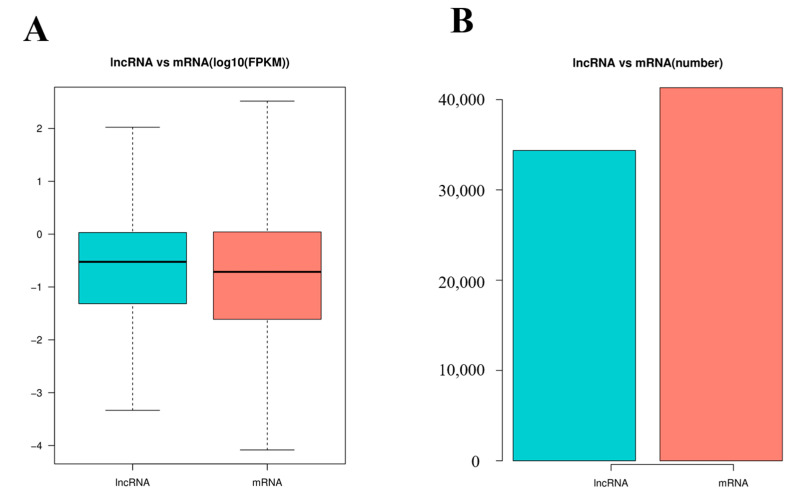
The expression levels and amounts of lncRNAs and mRNAs. (**A**) Boxplots of lncRNAs and mRNAs expression levels [with log10 FPKM (Fragments per Kilobase Million) method] in the control group and treatment group. (**B**) The numbers of lncRNAs and mRNAs in mouse spleens in the control group and treatment group.

**Figure 4 vaccines-11-00529-f004:**
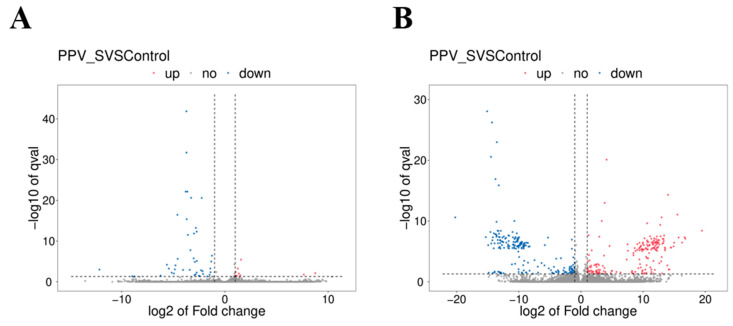
Volcano plot of the differential expression of mRNAs and lncRNAs in mouse spleens between the control group and treatment group. (**A**) Differential expression of mRNAs. The blue points denote significantly downregulated mRNAs, while the red points denote significantly upregulated mRNAs. (**B**) Differential expression of lncRNAs. The blue points denote significantly downregulated lncRNAs, while the red points denote significantly upregulated lncRNAs.

**Figure 5 vaccines-11-00529-f005:**
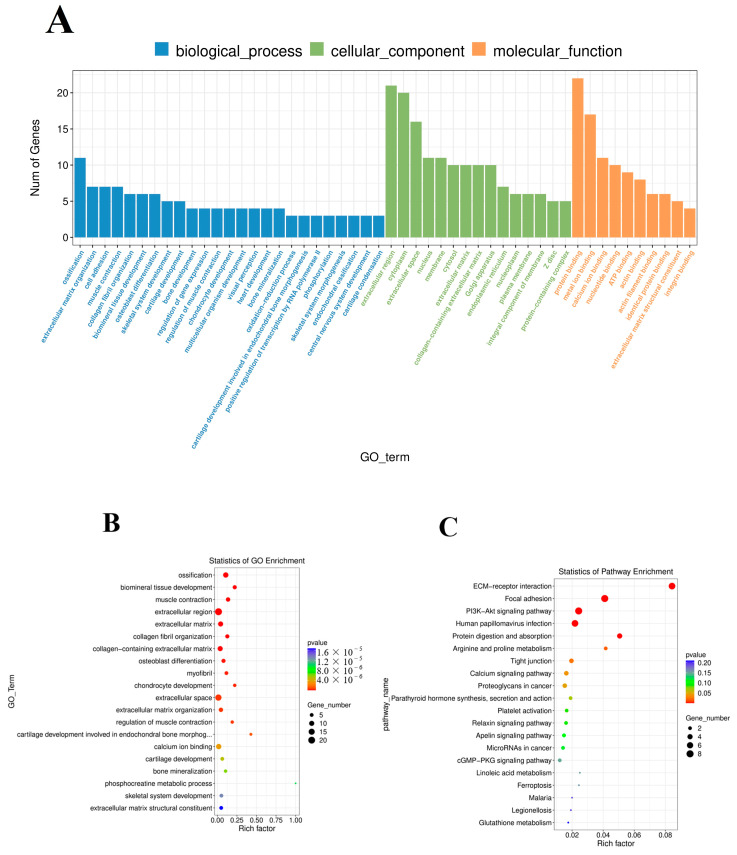
GO and KEGG analysis of differentially expressed mRNAs. (**A**) Histogram of GO enrichment of DE mRNAs. (**B**) Scatter plot of GO enrichment for DE mRNAs. (**C**) Scatter plot of KEGG enrichment for DE mRNAs.

**Figure 6 vaccines-11-00529-f006:**
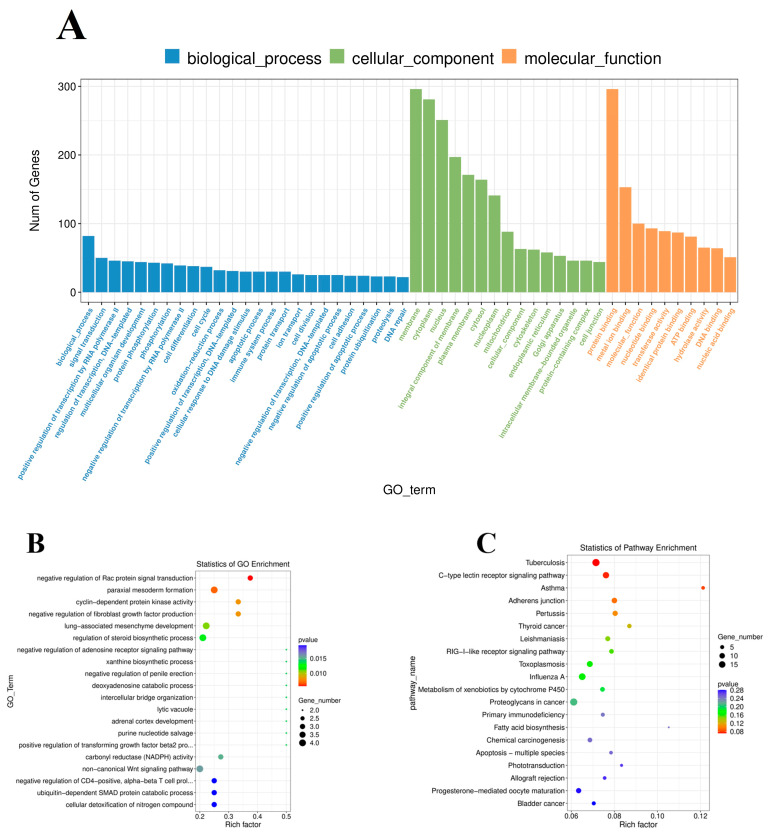
GO and KEGG analysis of differentially expressed lncRNAs. (**A**) Histogram of GO enrichment of DE lncRNAs. (**B**) Scatter plot of GO enrichment of DE lncRNAs. (**C**) Scatter plot of KEGG enrichment of DE lncRNAs.

**Figure 7 vaccines-11-00529-f007:**
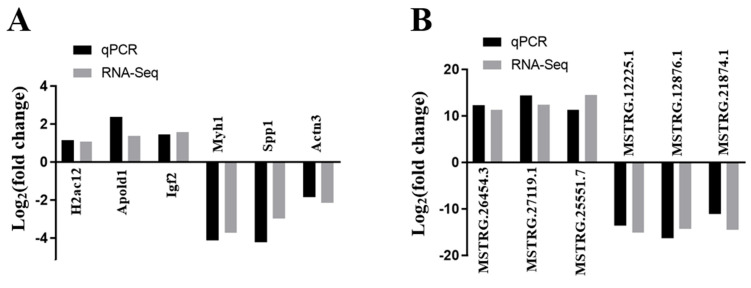
The validation of RNA−seq using qRT−PCR (n = 3). (**A**) qRT-PCR validation of six mRNAs. (**B**) qRT−PCR validation of six lncRNAs.

**Figure 8 vaccines-11-00529-f008:**
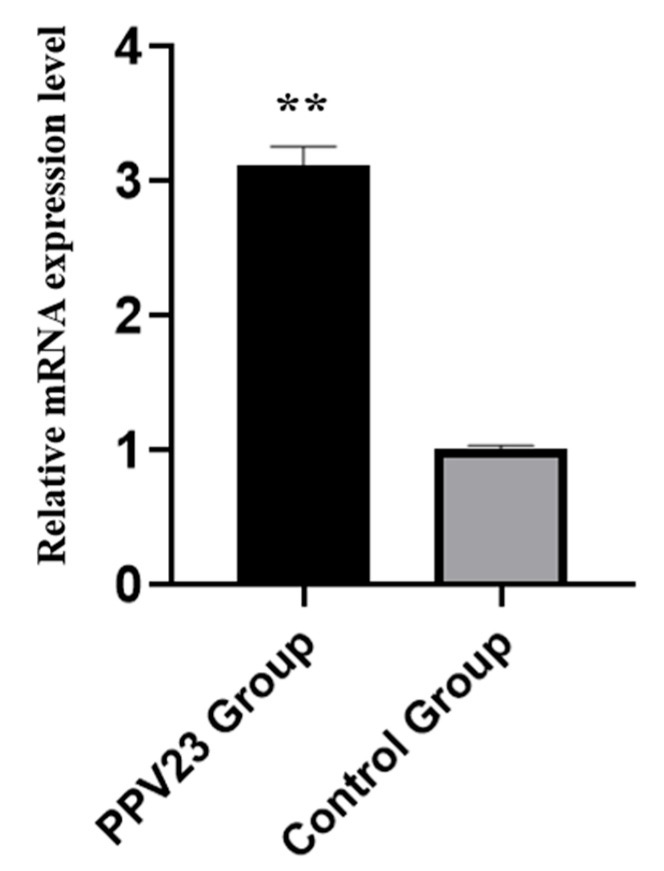
Relative mRNA expression levels of mouse Trim35 gene in spleens between the PPV23 treatment group and the control group (n = 3). ** indicates significant differences (*p* < 0.01).

**Table 1 vaccines-11-00529-t001:** Differentially expressed lncRNA-gene pairs between PPV23 and control group.

Gene Name	lncRNA Transcript Name	Cislocation (bp)	Pearson Correlation Coefficient
*St6galnac1*	MSTRG.5738.1	100 K	1
*Tmem106a*	MSTRG.5460.2	100 K	1
*St6galnac1*	ENSMUST00000156293	100 K	1
*St6galnac1*	ENSMUST00000147508	100 K	1
*Cdo1*	MSTRG.13054.1	100 K	−0.90
*Vamp8*	MSTRG.21907.5	1 K	−0.77
*Lin7b*	ENSMUST00000211098	100 K	−0.74
*Pcf11*	MSTRG.23880.5	10 K	−0.63

**Table 2 vaccines-11-00529-t002:** Co-enriched GO terms of DE lncRNA and DE mRNA.

GO Term	GO Function	*p*-Value
CD86 biosynthetic process	biological_process	0.03
CCR chemokine receptor binding	molecular_function	0.03
positive regulation of type IIa hypersensitivity	biological_process	0.05
mRNA cleavage factor complex	cellular_component	0.02
natural killer-cell-mediated cytotoxicity biological	biological_process	0.04

## Data Availability

The data presented in this study are available in supplementary material here.

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
