# Peer review of "Mechanisms of Immune-Related Long Non-Coding RNAs in Spleens of Mice Vaccinated with 23-Valent Pneumococcal Polysaccharide Vaccine (PPV23)"

_vaccines, 2023, doi:10.3390/vaccines11030529_

Round 1

Reviewer 1 Report

Estimated Authors,

I've asked to review this interesting study on the analysis of mechanisms of immune-related long non-coding RNAs in spleen of mice vaccinated with 23-valent pneumococcal polysaccharide vaccine (PPV23). In this study, Zhu et al. have injected a total of 12 mice (vs. 12 controls) with PPV23, with following analysis of the mRNA / lncRNA elicited by the injection. lncRNA are a type of RNA, generally defined as transcriptis more than 200 nucleotides that are not translated into protein. lncRNA is usually acknowledged as important regulators of the innate immunity (cfr. https://www.ncbi.nlm.nih.gov/pmc/articles/PMC6630897/) acting through the regulation of the expression of multiple inflammatory mediators and other responses, but are also emerging as an important regulator of biological process, such as chromatin remodeling, gene transcription, protein transport, and trafficking, eventually modulating the immunoglogulin response.

According the present results, a large number of mRNA (41321) and lncRNA (34375)s were identified, including 55 significantly differentially expressed (DE) mRNAs and 389 differentially expressed (DE) lncRNAs. Analyses indicated that the target genes of DE lncRNAs and DE mRNAs were seemingly related to T cell costimulation, contributing to the positive regulation of T cell differentiation, and intracellular biosyntetic pathways  indicating that the polysaccharide component antigens of PPV23 might activate cellular immune response during PPV23 immunization process. 

In other words, the present reviewer has identified this article as both interesting and appropriately designed and written. Some minor amendments are (from my point of view) required, and namely:

1) in the introduction , Authors should share some further details on the presumptive role of lncRNA in the immune system and on their potential association with mRNA synthesis (cfr. https://www.ncbi.nlm.nih.gov/pmc/articles/PMC4451983/ https://www.ncbi.nlm.nih.gov/pmc/articles/PMC6630897)

2) explain, please, why did you compare the overall synthesis of mRNA and lncRNA, as (at least according to my previous understanding of this topic) the crude number of sequences does not represent a direct index of their biological role

3) Figure 5A should be refined by increasing the size of the captions

4) please include the explanation of the meaning of lncRNA acronym in the abstract

Author Response

1) in the introduction , Authors should share some further details on the presumptive role of lncRNA in the immune system and on their potential association with mRNA synthesis (cfr. https://www.ncbi.nlm.nih.gov/pmc/articles/PMC4451983/ https://www.ncbi.nlm.nih.gov/pmc/articles/PMC6630897)

Some further details on the presumptive role of lncRNA in the immune system and on their potential association with mRNA synthesis have been shared in the introduction in line 90-105 & 110-125.

2) explain, please, why did you compare the overall synthesis of mRNA and lncRNA, as (at least according to my previous understanding of this topic) the crude number of sequences does not represent a direct index of their biological role

Yes, the crude number of sequences does not represent a direct index of their biological role. LncRNAs are generally shorter in length, have fewer, but longer exons and are expressed at lower levels compared to mRNAs. The purpose we compared the overall synthesis of mRNA and lncRNA in this study was to confirm the quantitative difference between mRNA and lncRNA, and compare with other studies to see the reliability of our sequencing and analysis results.

3) Figure 5A should be refined by increasing the size of the captions

Because the picture is large and there are too many items, the enlarged picture has been uploaded to the attachment for viewing

4) please include the explanation of the meaning of lncRNA acronym in the abstract

It has been added in the abstract in line 23.

Reviewer 2 Report

This is an important study. My comments are directed to helping other researchers benefit by your work.

Introduction

“Many previous studies have 66 shown that Pneumococcal polysaccharide vaccine can induce 67 non-T cell-dependent humoral immune response, so it is more 68 suitable for the susceptible population over 2 years old or over 50 69 years old (Jcakson et al., 2007).”

[Please provide data. Patient susceptibility and defences are key concerns of your vaccine study]

“played key roles in the regulation of mRNA expression 87 by affecting the transcription process or recruiting epigenetic 88 complexes directly”

[Please state how mRNA expression affects the transcription process and which epigenetic complexes are recruited and how these complexes are involved in the infectious process]

“More 89 specifically, lncRNAs could recruit some transcription factors to 90 genomic DNA, destabilizing messenger RNA (mRNA) and 91 segregating micro-RNAs (miRNA) (Fatica, A & Bozzoni, 92 2014). As a consequence, the genetic molecular mechanisms of 93 cell proliferation and differentiation, regulation of cell cycle, 94 dosage compensation and epigenetics were all participated in the 95 protein inhibition by miRNAs titration or by binding of lncRNAs 96 to proteins or to miRNAs (Harinarayanan et al., 2018). On the 97 basis of molecular mechanisms of lncRNAs, they can also act as Vaccines 2021, 9, x FOR PEER REVIEW 3 of 17 98 guides, decoys, signals and scaffolds during the biological 99 process (Nojima & Proudfoot, 2022). Long non-coding RNAs also 100 have been identified as key regulatory factors in a number of 101 biological processes including immune, but the regulatory 102 mechanisms of most lncRNAs in immune processes still remain 103 largely unknown (Wu et al., 2022).”

[identifying these complex processes is the key problem you have set yourselves. Please provide as much detailed information as you can to show what is known and what is not known, the  hypotheses based on the literature that you are testing, and the results of testing the hypotheses. I realise that your work is of an exploratory nature but stating hypotheses is key]

“2.3. Quality Control and Mapping 160 The low-quality reads of the sequencing data, including low 161 quality bases, undetermined bases and adaptor contamination, 162 were removed by Cutadapt. FastQC was applied to verify the 163 sequence quality. Hisat2 (Kim et al., 2015) and Bowtie2 164 (Langmead et al., 2012) were then used to map the obtained reads 165 to the genome of mousee in NCBI (Mus musculus, assembly 166 GRCm39). StringTie was applied to assemble the mapped reads. 167 All the transcripts from mouse spleen were combined to 168 reconstruct a comprehensive transcriptome by a Perl script. 169 Ballgown (Frazee et al., 2015) and StringTie (Pertea et al., 2015) 170 were applied to estimate the expression levels of all the 171 transcripts after the final transcriptome were generated.”

[Please describe the problems you encountered and how you solved them in this complex process which deletes information]

“The previous studies 526 reported that all polysaccharides belong to T cell-independent 527 type 2 (TI-2) antigens, after entering the body, they bind to 528 antigen recognition receptors on the surface of B cells, namely 529 surface immunoglobulin molecules, crosslinks and activates B 530 cells, causing a series of downstream changes, so that B cells 531 differentiate into plasma cells that can secrete antibodies 532 (Masomian et al., 2020). However, with the above evidence 533 combined with the results of this study, we speculate that T cells 534 might involve in the immunization process of PPV23. The 535 specific molecular mechanism of how do T cells involve in the 536 immune process caused by capsular polysaccharide of PPV23 537 still need further identified and intensive study.”

“The 545 lncRNA MSTRG.9127 might play an important regulatory role in 546 immunologic process treated with PPV23 by affecting its 547 potential target gene Trim35. Therefore, our study might lay a 548 foundation for the immune mechanism of polysaccharide 549 vaccines and the development of future vaccines”

[Can you make conclusions from your data other than “speculating” and ”might.”?]

The English text is very readable but needs many minor corrections by a native English speaker.

Author Response

1) [Please provide data. Patient susceptibility and defences are key concerns of your vaccine study]

The data has been added in line 68- 72.

2) [Please state how mRNA expression affects the transcription process and which epigenetic complexes are recruited and how these complexes are involved in the infectious process]

They were stated in line 90-125.

3) [identifying these complex processes is the key problem you have set yourselves. Please provide as much detailed information as you can to show what is known and what is not known, the  hypotheses based on the literature that you are testing, and the results of testing the hypotheses. I realise that your work is of an exploratory nature but stating hypotheses is key]

This is a very valuable comment, which is very helpful to the overall quality of the manuscript. We try to provide much detailed information and attempt to make it clear. The supplemental information was added in line 129-137.

4) [Please describe the problems you encountered and how you solved them in this complex process which deletes information]

To be honest, it was really a complicated process and there were many problems, big and small. The first is to learn a variety of bioinformatics software and data processing software. The first step after obtaining the transcriptome data (.fastq file) is to control the quality of the raw data. The purpose of quality control is to comprehensively check the quality of the original data, including base quality assessment, GC content inspection, N-base quantity assessment, TCGA base distribution, k-mer quantity inspection, etc. The software FastQC was applied to verify the sequence quality. Then the softwares Hisat2, Bowtie2 and StringTie were used to map the obtained reads to the mouse genome. The initial analysis of the data was done with the help of laboratory colleagues with experience in transcriptome analysis. Although there were a lot of small problems along the way, but finally with the help of the sequencing company, we completed the difficult step.

5)  [Can you make conclusions from your data other than “speculating” and ”might.”?]

The conclusion has been modified in line 587-590.

6) The English text is very readable but needs many minor corrections by a native English speaker.

The manuscript has been modified by a native English speaker.

Reviewer 3 Report

The manuscript is an interesting analysis of the response to the pneumococcal polysaccharide vaccine PPV23. There is an important amount of data, analyzed accordingly. The data is in the supplementary files as well as the figures and seem to correspond. 

There are some minor details that need to be included, how the vaccine was administered, back, tight, ip? A polysaccharide vaccine should enhance more IgM than IgG; any explanation for this phenomenon? The IgM and IgA values are very similar in the vaccinated group. Do the authors have any explanation? 

According to figure 5, ECM and focal adhesion are critical; however, it is not properly discussed along with asthma and probably COPD.

In overall, a good interesting manuscript

Author Response

  • There are some minor details that need to be included, how the vaccine was administered, back, tight, ip?

The vaccine was injected intramuscularly into the lateral thigh muscle of mice, and it was added in line 170-171.

  • A polysaccharide vaccine should enhance more IgM than IgG; any explanation for this phenomenon? The IgM and IgA values are very similar in the vaccinated group. Do the authors have any explanation? 

For this phenomenon, we have not done in-depth molecular experiments to explain this. We speculated the possible reason was that the time of serum collection. After the antigen enters the body for the first time, plasma cells will be produced and antibodies will be synthesized and secreted after a certain incubation period. The first was IgM, but this antibody is short-lived and disappears quickly, lasting from days to weeks in the blood, and then IgG produced next. The time that we collected the serum might be the time that the IgM was producing its upsurge or peak. nonetheless, The absolute value of igg concentration was higher than that of IgM and IgA, although the trend of increase was more obvious in significance analysis.

  • According to figure 5, ECM and focal adhesion are critical; however, it is not properly discussed along with asthma and probably COPD.

This is a very good comment. The discussion of “ECM and focal adhesion” have been added in line 500-512.

Round 2

Reviewer 2 Report

Thanks to the authors for responding to the reviewer's comments about this complex study.

[The changes need proof reading please. For example the final word date should be data].